# Learning Cephalometric Landmarks for Diagnostic Features Using Regression Trees

**DOI:** 10.3390/bioengineering9110617

**Published:** 2022-10-27

**Authors:** Sameera Suhail, Kayla Harris, Gaurav Sinha, Maayan Schmidt, Sujala Durgekar, Shivam Mehta, Madhur Upadhyay

**Affiliations:** 1Department of Engineering Technologies, Swinburne University of Technology, Hawthorn, VIC 3122, Australia; 2Private Practice, Lakewood, CO 80226, USA; 3Departments of Computer Science & Statistics, University of British Columbia (Alumni), Vancouver, BC V6T1Z4, Canada; 4School of Dental Medicine, University of Connecticut Health, Farmington, CT 06030, USA; 5Department of Orthodontics, KLES’ Institute of Dental Sciences, Bangalore 560022, India; 6Department of Developmental Sciences/Orthodontics, Marquette University, Milwaukee, WI 53202, USA; 7Division of Orthodontics, University of Connecticut Health, Farmington, CT 06030, USA

**Keywords:** cephalograms, anatomical landmarks, machine learning, regression trees, orthodontics

## Abstract

Lateral cephalograms provide important information regarding dental, skeletal, and soft-tissue parameters that are critical for orthodontic diagnosis and treatment planning. Several machine learning methods have previously been used for the automated localization of diagnostically relevant landmarks on lateral cephalograms. In this study, we applied an ensemble of regression trees to solve this problem. We found that despite the limited size of manually labeled images, we can improve the performance of landmark detection by augmenting the training set using a battery of simple image transforms. We further demonstrated the calculation of second-order features encoding the relative locations of landmarks, which are diagnostically more important than individual landmarks.

## 1. Introduction

Lateral cephalometric radiographs have played a central role in the diagnosis of malocclusions. Diagnosis proceeds by locating certain important anatomical landmarks on the cephalograms and evaluating the higher-order features obtained from the relative placement of these points. The manual localization of landmarks is a tedious and time-consuming task, which even in the case of experienced orthodontists can take approximately 10–15 min/radiograph [1]. In addition, significant discrepancies may arise between different evaluators. Since orthodontic diagnosis and associated treatment procedures are sensitive to the accurate estimation of landmarks, inconsistencies in landmark identification can potentially have deleterious effects on diagnoses and treatment outcomes. Automated landmark detection on lateral cephalometric radiographs using machine learning techniques can help an orthodontist locate landmarks instantaneously while avoiding subjective inconsistencies [2,3,4,5,6]. Previously, Lindner et al. attempted to predict landmarks in lateral cephalograms and used them in the classification of skeletal malformations [7]. Similarly, Wang et al. used multiscale decision tree regression voting using scale-invariant [8] patch features for landmark detection in cephalometric X-rays to calculate certain clinical parameters [9]. There have also been attempts to locate cephalometric landmarks without the application of machine learning. Grau et al. conducted landmark detection using a line detection module followed by the application of pattern matching techniques [6]. The Automatic Cephalometric X-Ray Landmark Detection Challenge, held at the IEEE (Institute of Electrical and Electronics Engineers) International Symposium on Biomedical Imaging 2014, saw two machine learning algorithms perform the best for different precision ranges [1]. Ibragimov et al. conducted landmark detection by applying a combination of game theory and random forests that showed better performance for smaller precision ranges [10,11]. Vandaele et al., on the other hand, applied an extremely randomized tree-based approach that performed better on a larger precision range [12]. A form of data augmentation was applied by Oh et al. that employed a local feature perturbator on the local clues and inputs of differently perturbed images in each epoch for the convolutional neural network while training [13].

The aim of automatic landmark detection in cephalometric X-rays is to save the time spent on manually labeling landmarks on a cephalogram and to improve the measurement accuracy since both the interoperator and intraoperator variability of measurement errors can be eliminated [14,15,16]. This aids in orthodontic diagnosis and treatment planning. The performance of machine learning algorithms in multiple domains has increased substantially in recent years. This progress is a result of advances in computational capacity and associated methods that are well-suited to process larger datasets. For supervised learning tasks, such as the landmark detection of medical images, the availability of correctly labeled data is a key resource. However, accurate and medically precise labeling can only be performed by trained personnel. As labeled data can be considered a precious and expensive resource, data augmentation methods can be beneficial in reducing the need for larger amounts of labeled data.

Therefore, this study was undertaken with the objective to automate the detection of these landmarks using machine learning and then compute diagnostic features using the landmark locations. Furthermore, in this study, a number of data augmentation methods and their performances were evaluated. The relevant question that this research aimed to answer is whether we can artificially increase the size of the training data while keeping the labeling accurate with the different data augmentation methods. Therefore, the performance of the proposed model was calculated based on both the predicted landmarks and the features stemming from the landmarks.

## 2. Materials and Methods

### 2.1. Dataset

The dataset comprised 362 lateral cephalometric radiographs, each labeled with 26 landmarks. Based on a previous study, a sample of 300 cephalograms would be considered sufficient for data augmentation. In our study, 375 cephalometric images were obtained to account for exclusions due to poor image quality, distortion, an incomplete region of interest of the skull, etc. Out of the 375 images, 362 images were included in the study. The dataset did not require institutional review board (IRB) approval since the lateral cephalometric radiographs were anonymized and no demographic or other identifiable information was obtained. Figure 1 shows the landmarks for a sample lateral cephalometric radiograph. Table 1 describes each landmark in further detail. These landmarks were marked on all 362 cephalometric X-rays by two trained orthodontists (KH and MU). Together, the orthodontists had more than 20 years of orthodontic experience.

The landmarks were then used to calculate a number of angular features that are diagnostically relevant for orthodontic treatment. Figure 2 shows the angles on a sample cephalometric X-ray.

### 2.2. Ensemble of Regression Trees

In order to solve this nontrivial problem of pinpointing the various landmarks (Table 1) using only the X-ray image, we employed the approach of averaging over a large number of relatively simple predictors. This is akin to computing the average opinion of a cohort of orthodontic students rather than depending on the expertise of a single senior and experienced orthodontist. This approach has two advantages. Firstly, a generic and relatively simple algorithm can be used repeatedly rather than crafting a very sophisticated algorithm. Secondly, averaging over multiple predictors avoids the idiosyncrasies that might be introduced in a sophisticated predictor. In machine learning, this technique is called ensemble learning and is considered a robust approach. Specifically, we employed a large number (ensemble) of regression trees to estimate the landmark locations on the cephalometric X-rays. Regression refers to the prediction of a numerical value (here, the x and y coordinates of a landmark). These predictions were calculated using the image intensity values at various points in the cephalometric X-rays. Regression trees can be thought of as a sequence of successive yes or no questions that narrow down the range of landmark locations after each question. Just as a tree branches, our algorithm also branches to two nodes at each stage. After the last question, we arrive at the most likely position of the landmark.

This algorithm was first developed for facial alignment in 2014 [17]. Its implementation is available in the dlib [18] library for Python [19]. Each regressor in the cascade is trained using the gradient tree boosting algorithm with a sum of square error loss. Each tree comprising the ensemble is limited to a certain tree depth parameter. The number of successive ensembles specifies the cascade depth, whose combined results generally give better accuracy than using only a single-stage ensemble. The learning parameters that we experimented with in the present study are nu, the cascade depth, the tree depth, and the oversampling amount. The definitions of these parameters are given in Table 2.

Of the total number of cephalograms, 70% (253 images) were used for training, and the rest (109 images) were used as a test set. The images in the training set were shuffled to ensure randomness in the order they were processed by the model being trained.

### 2.3. Transformations

Since collecting and annotating X-rays is costly, we attempted to artificially increase the number of images by applying different transforms to the X-rays: zoom, horizontal shift, vertical shift, shear, rotate, and elastic transform. This is a standard machine learning method to augment the training data size for more accurate predictions. We gradually increased the augmented training set by transforming each image up to six times. The parameters of the transformations are given in Table 3.

## 3. Results

The distance between the true and predicted positions of a landmark is the measured error that was used to determine the accuracy of the model. This is the most apt measure of performance, as the focus is to estimate the landmark position as close to its true position as possible. Additionally, the model was also assessed based on the directionality of the errors. This takes into account whether the model is prone to predicting the landmark position equally in all directions away from the true position or if there is a bias towards a certain direction. This error measurement is necessary to gain confidence in the predictions made by our model.

The learning parameters of the model were nu, the tree depth (F), the oversampling amount (R/N), and the cascade depth (T). We were able to find a set of intermediate parameters, such as nu = 0.2, tree depth = 1, and oversampling amount = 20, with the lowest test error (Figure 3a). The error continued to decrease with increased complexity (Figure 3b). The training time complexity was proportional to F*R*T. This is demonstrated in the training times in Figure 3c.

To arrive at the highest possible accuracy on the test set, the learning parameters were varied over a certain range and the model was trained for each such variation of the parameters. The resulting test set error, training set error, and training times were plotted against the learning parameters. The errors were in pixels averaged over all 26 landmarks over all images in the corresponding set. The experiment gave the optimal values of the learning parameters for the currently available training data.

### 3.1. Novelty Introduced by Data Augmentation Methods

It was expected that increasing the number of training data samples would result in higher accuracy on unseen images. Therefore, it was desirable to increase the size of the training set while keeping the expert labeling costs of the X-rays to a minimum. Artificially augmenting the training data using different transforms and feeding those images to the model being trained may make the final trained model more robust. Figure 4a shows the transforms (zooming, height shifting, width shifting, shearing, rotating, and nonlinear elastic transformation) used on a sample image. We assessed the “novelty” of each transformation from the perspective of both the training and the performance of the models.

For a model trained on original images only, Figure 4b shows the test errors obtained when predicting the landmarks of transformed images. Obviously, the error on the untransformed trained images was negligible. After transforming the training images, the model was not able to perfectly predict the landmark positions, implying that these transformed images appear novel to the model. We see that the rotation and shear transformations introduced the most novelty. The errors for the elastic transformation were higher by about one order of magnitude compared to the rest of the transforms.

A good transformation from the perspective of data augmentation would be one that makes transformed images appear only as different as any other unseen image. Therefore, we also compared the performance of the model on unseen images vs. the transformations of the unseen images. Figure 4c shows the prediction errors on the transformed and untransformed unseen images. All the transforms resulted in prediction performances similar to those of the untransformed images, apart from the elastic transformation.

We concluded from this analysis that relatively simple transforms introduce novelty into the images to make them appear different from the original images without distorting the images to the extent of catastrophic performance (as seen in the elastic transformation). Subsequent analyses in this study therefore included only zoom, shifts, rotation, and shearing and excluded the elastic transformation.

### 3.2. Performance Gains from Data Augmentation

As we have seen, even though the augmented images were only slightly different compared to the original images they were sourced from, they were still able to provide some diversity to the model. This can be seen in Figure 5, where successively adding more images to the training set led to a drop in error. The drop was more significant when the added images were new original images instead of transformed versions of the 50 original images. For each image, the transformation parameters were randomly chosen from within a range. The range was chosen based on the transformation parameters used in Figure 4. Therefore, the ranges were 20%, −70 to +70 pixels, and 15 degrees for zoom, shifting, and shearing, respectively.

Once the model was finalized, its accuracy was separately evaluated for each landmark. Figure 6 shows the distribution of the prediction errors in pixels. Figure 7 shows the directionality of these errors in degrees. The errors were symmetrically distributed for almost all landmarks.

These raw landmarks provide the basis for the selection of diagnostically important features. Seven angular features were identified and analyzed based on the identified landmarks. The distribution of the prediction errors of these features is shown in Figure 8.

## 4. Discussion

This study evaluated the use of the ensemble of regression tree method for cephalometric landmark detection. This method has mostly been used for facial landmark detection [20,21,22,23]. This method relies on iteratively refining the location of each landmark using both the local pixel features and the estimates of the locations of other landmarks. The diagnostically important landmarks are located farther apart throughout the image compared to the clustered locations of facial landmarks as they are generally defined.

In the current study, each cephalometric X-ray had 26 landmarks. The landmarks represented distinct points distributed throughout the images rather than defining a contour. The original paper and subsequent works, however, used this algorithm mainly for facial landmark detection, where the landmarks were clustered together and formed a contour around facial features.

Even though the number of images for both training and testing the model were just about sufficient for machine learning purposes, we obtained satisfactory performance with our method. This could be owing to the use of data transforms that augmented our dataset, artificially increasing the training set. This study demonstrates the usefulness of image transformations in terms of the novelty introduced by transformed images compared to real unseen images. The trained model’s performance was satisfactory, not just in terms of how close the predicted landmarks were to the true landmarks but also in how the predicted landmarks were distributed around the true landmarks for different images.

For applications in clinical orthodontics, the direction of the landmark prediction is important for the outcome assessment. It was observed that the prediction made by the current method for the landmark glabella did not sway outward or inward from the skull periphery, as shown in Figure 7. This is intuitively consistent with the expectation of a human orthodontist. Consistent identification of the glabella ensures higher accuracy when measuring the soft-tissue convexity of the profile, which is measured by the lines joining the three landmarks glabella, subnasale, and soft tissue pogonion [24]. The current model not only predicted the individual landmarks with appreciable accuracy but also preserved the angular relationships between these landmarks, which is crucial for a correct diagnosis (Figure 8). Specifically, the standard deviation of the errors for the angle ANB [A point]-[Nasion]-[B point] were within 4 degrees. This is similar to the research findings of Hunag et al. on the analysis of cephalometric measurements using artificial-intelligence-based algorithms [25].

The current study also shows how the identification of landmarks affects the different angles used for orthodontic diagnosis. SNA relates the maxilla to the cranial base. SNB measures the relation of the mandible to the cranial base [26,27]. ANB measures the difference between SNA and SNB. The prediction of SNA, SNB, and ANB showed some error as per the current model (Figure 8). However, interestingly, the angle ANB demonstrated a smaller error than both SNA and SNB. ANB is the net output obtained by the difference between SNA and SNB. The smaller difference of ANB may show the error of each SNA and SNB in each sample. This could be due to the directionality of the errors in SNA and SNB such that the errors lined up antagonistically with each other and not synergistically, resulting in a lower net ANB error.

In recent years, a number of artificial-intelligence-based algorithms have been studied to evaluate the performance of machine learning on cephalometric radiographs. In this study, augmentation methods were used to identify similar cephalometric landmarks that were used in previously published artificial intelligence studies [28,29,30]. When training a machine learning algorithm, increasing the number of training data samples results in a higher accuracy on unseen images [31]. However, increasing the amount of labeled data is a challenge in orthodontics, as labeling cephalometric radiographs requires input from orthodontists. This results in increased expert labeling costs and requires substantial expert time. Both these factors can limit the amount of labeled data available. Therefore, it is desirable to increase the size of the training set while keeping the expert labeling costs to a minimum. This study showed that augmentation methods can be successfully used to increase the training data sample by utilizing an ensemble regression tree method for the analysis of cephalometric radiographs.

There were certain limitations to the study such as the sample size used in the AI algorithm. However, even with that limitation, the ensemble regression tree algorithm performed well in the identification of cephalometric landmarks when used with augmentation methods. This study demonstrates that the different types of image transformations, such as original, zoom, H-shift, W-shift, shear, and rotation, are useful for improving the accuracy of the artificial intelligence model for the prediction of landmarks used for orthodontic diagnosis in lateral cephalograms. The elastic transform, despite being more complicated, did not fare so well. One possible reason may be the unrealistic nature of the images generated by the elastic transform.

## 5. Conclusions

Ensemble of regression tree proved to be a reliable approach for the automated identification of cephalometric landmarks.Augmentation methods could be used to artificially increase the training set size and consequently improve the performance of the machine learning model.Simple image transformations such as original, zoom, H-shift, W-shift, shear, and rotation worked well to introduce novelty to the model.The elastic transformation did not perform well as a method of augmentation for introducing novelty to the sample.

## 6. Patents

Upadhyay et al. Artificial Intelligence (AI) based Decision-Making Model for Orthodontic Diagnosis and Treatment. Pub.No.:US2021/0196428A1. https://patents.google.com/patent/US20210196428A1/en?inventor=madhur+upadhyay&oq=madhur+upadhyay, 25 September 2022.

## Figures and Tables

**Figure 1 bioengineering-09-00617-f001:**
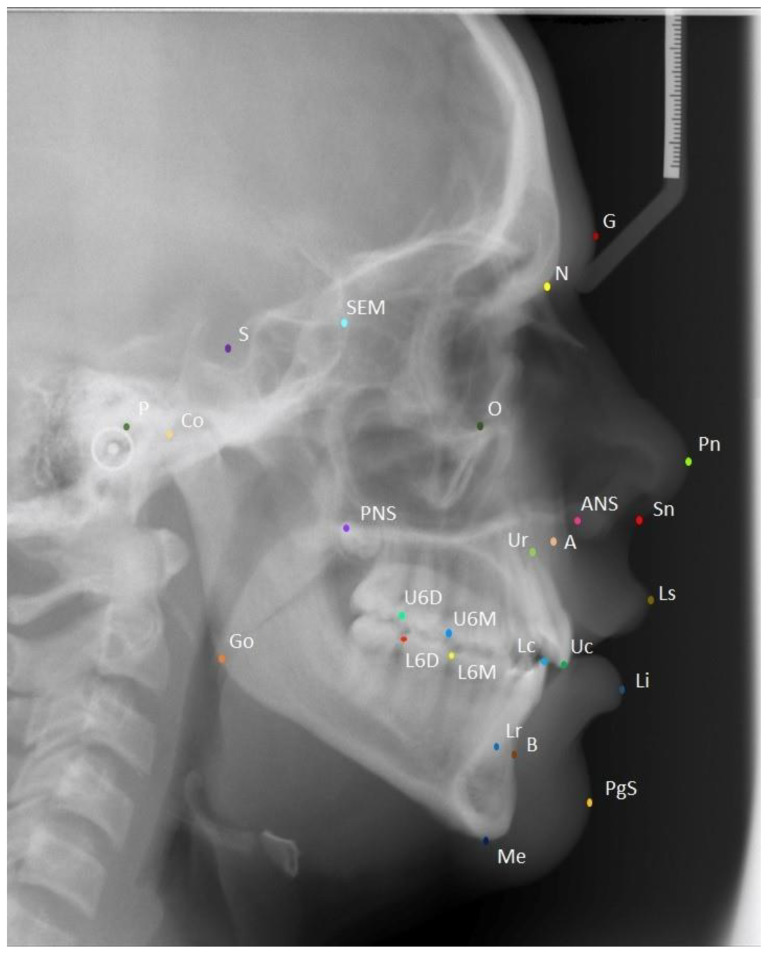
A sample cephalometric X-ray showing the true positions of the 26 cephalometric landmarks.

**Figure 2 bioengineering-09-00617-f002:**
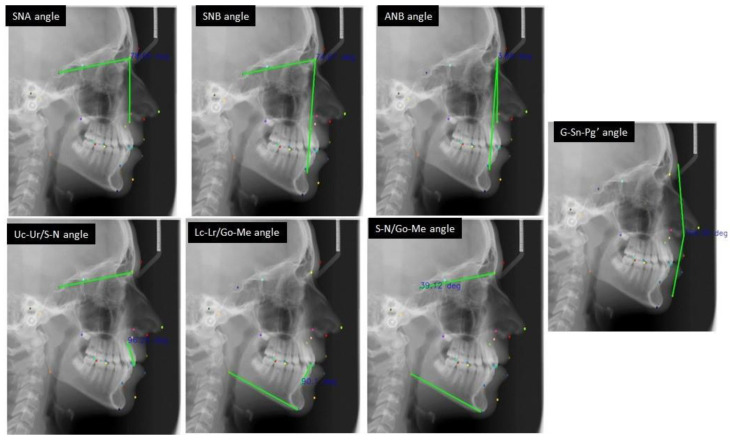
The seven angular measurements based on the cephalometric landmarks.

**Figure 3 bioengineering-09-00617-f003:**
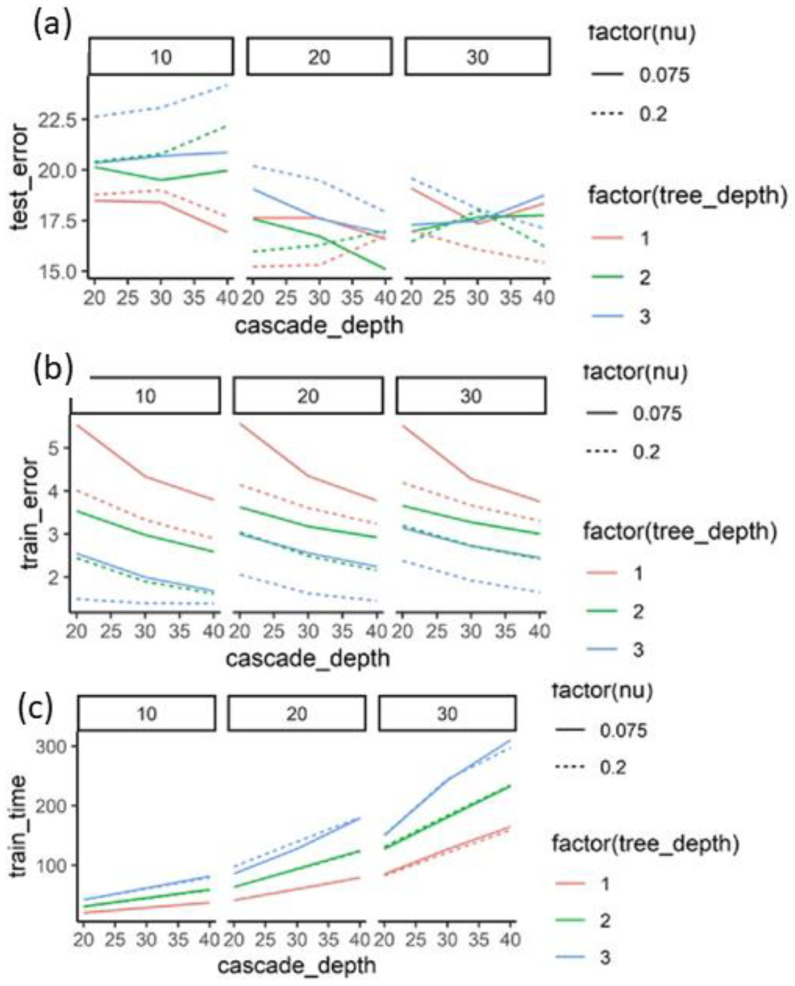
Effects of learning parameters on performance. (**a**) Mean distance between the predicted and labeled landmarks in pixels plotted on the y axis for the test images. (**b**) Mean distance between the predicted and labeled landmarks in pixels plotted on the y axis for the train images. (**c**) Training time in seconds for prediction models trained with different learning parameters.

**Figure 4 bioengineering-09-00617-f004:**
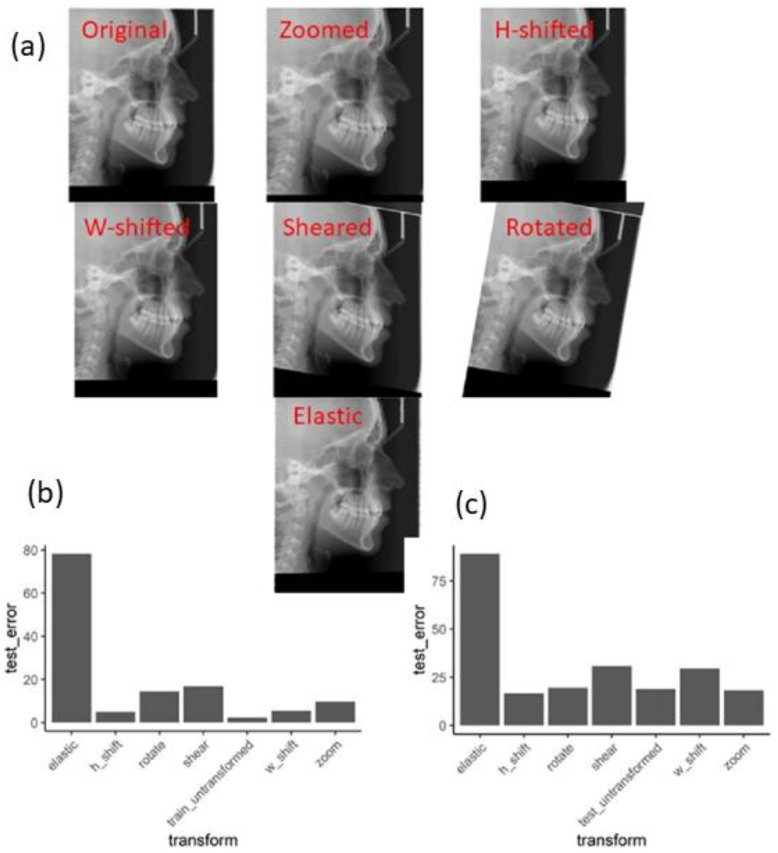
Evaluating different data augmentation transforms. (**a**) Sample images showing the results of image transformations. (**b**) A single model was trained on a set of 253 original (untransformed) images. The same training images were altered using different transforms. The performance of the original model was evaluated on these transformed images. The errors on the transformed images (train_untransformed) were of a similar magnitude to those on the real unseen images (test_untransformed), thus confirming that the transformations added an appropriate amount of novelty to the images. (**c**) The performance of the original model was evaluated on different transformations of new unseen (test) images. The transformations did not overly distort the images since the performance on the transformed images was similar to the performance on untransformed images.

**Figure 5 bioengineering-09-00617-f005:**
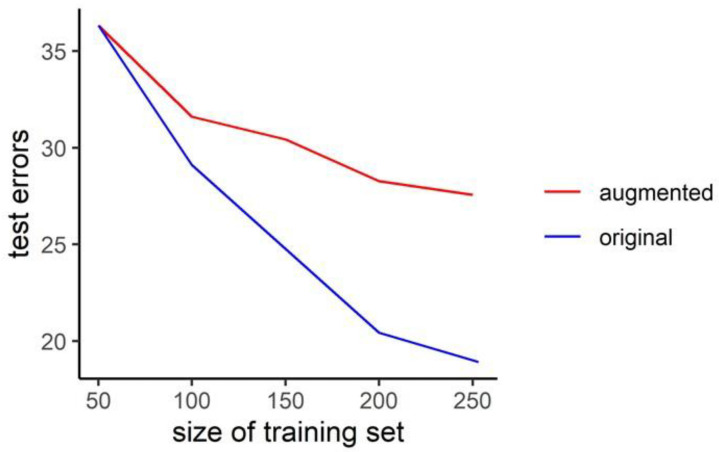
Gain in model performance by data augmentation. Starting from 50 untransformed images as a training set, the model was successively trained on larger datasets by adding either unseen images (blue line) or transformations of the same starting set (red line). The test error continued to decline with larger datasets, with higher gains from unseen images, but adding augmented images also led to a gain of at least 50% compared to the original images.

**Figure 6 bioengineering-09-00617-f006:**
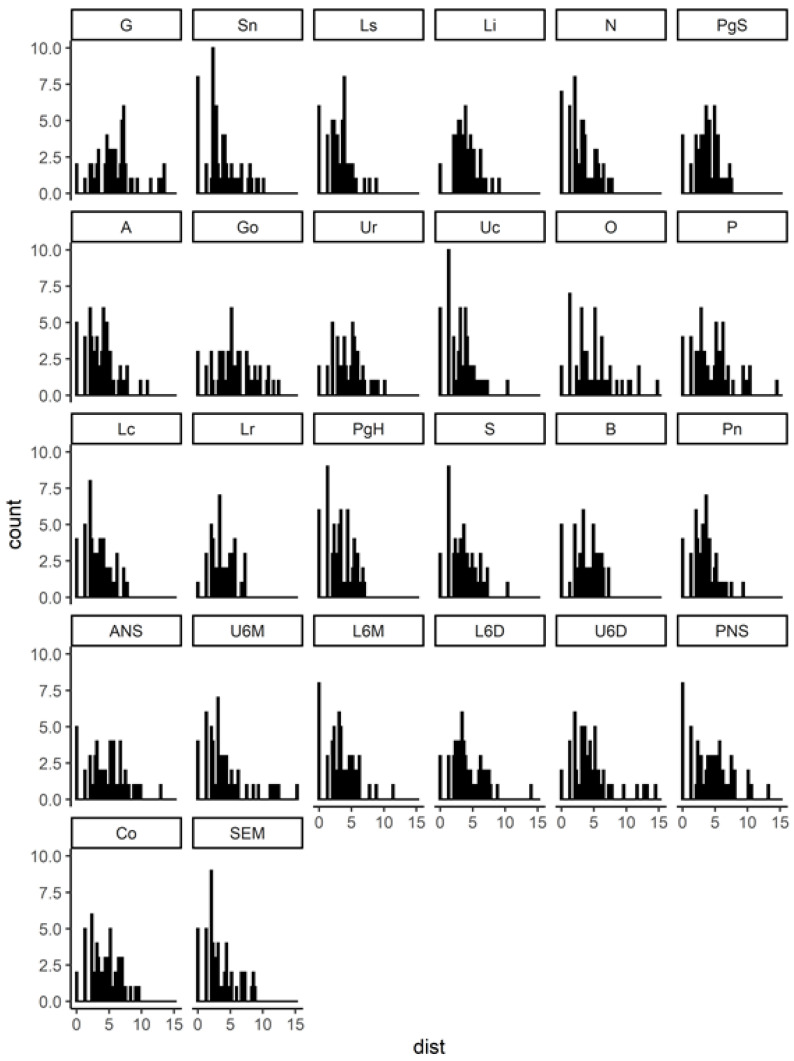
Prediction accuracy for each landmark. Each panel shows the distribution of the magnitude of prediction error in pixels (*x*-axis) for each of the 26 landmarks.

**Figure 7 bioengineering-09-00617-f007:**
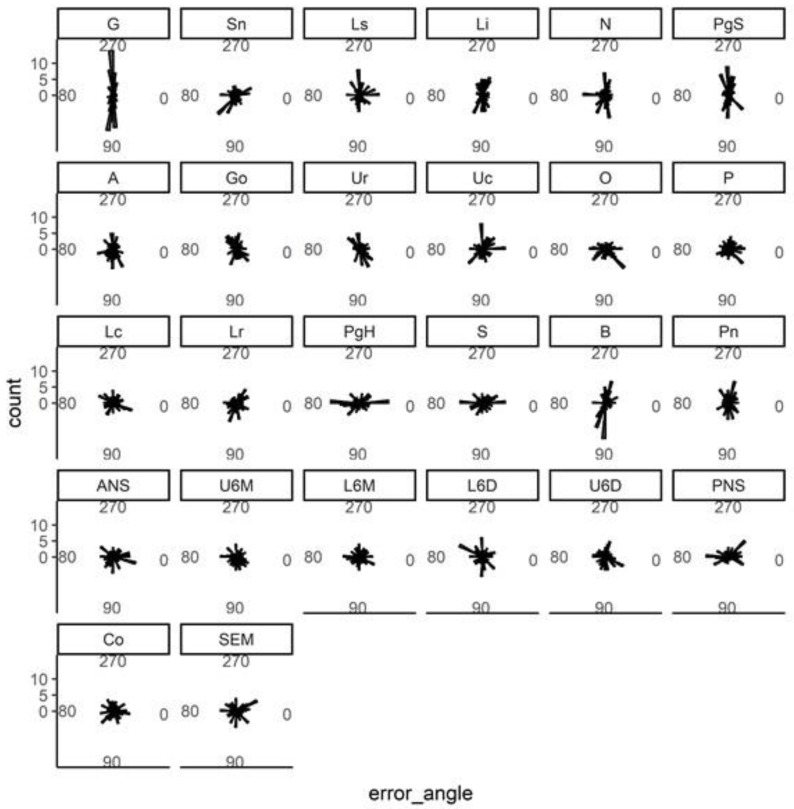
Directionality of the prediction errors of each landmark. Each panel shows the angular distribution of the prediction error in degrees (*x*-axis) for each of the 26 landmarks.

**Figure 8 bioengineering-09-00617-f008:**
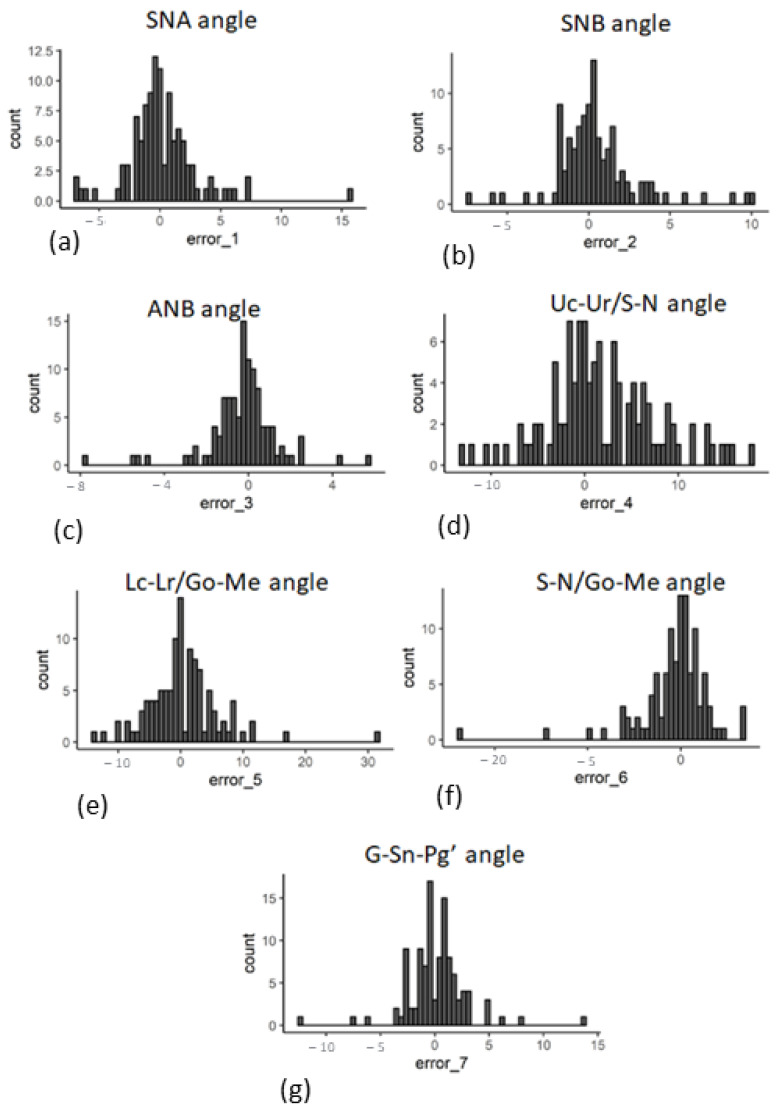
Error distribution in predicting the angular features. The distribution of the magnitude of the prediction error in degrees (*x*-axis) is shown for (**a**) SNA, (**b**) SNB, (**c**) ANB, (**d**) Uc-UR/S-N, (**e**) Lc-Lr/Go-Me, (**f**) S-N/Go-Me, and (**g**) G-Sn-Pg’ angle.

**Table 1 bioengineering-09-00617-t001:** Description of the landmarks used in the study.

Landmark	Description
S (Sella)	The geometric center point of the pituitary fossa.
P (Porion)	The most superior point on the external auditory meatus.
Go (Gonion)	Most posterior and inferior point on the curvature of the angle of the mandible.
N (Nasion)	The most anterior point on the frontonasal suture.
O (Orbitale)	The lowest point on the inferior rim of the orbit.
A (Point A)	The deepest point on the bony concavity between the ANS and supradentale.
B (Point b)	The deepest point on the bony concavity between the pogonion and infradentale.
Me (Menton)	The most inferior point on the hard tissue chin.
G (Glabella)	The most anterior point on the forehead.
Sn (Subnasale)	The junction of the nose and upper lip.
Ls (Labrale superius)	The most prominent point on the upper lip.
Li (Labrale inferius)	The most prominent point on the lower lip.
PgS (Soft tissue pogonion)	The most prominent point on the soft tissue chin.
Ur	The root tip of the upper incisor.
Uc	The crown tip of the upper incisor.
Lc	The crown tip of the lower incisor.
Lr	The root tip of the lower incisor.
ANS (Anterior nasal spine)	Anterior tip of the nasal spine.
PNS (Posterior nasal spine)	Posterior tip of the nasal spine.
U6M Upper first molar mesial tip	Most prominent point on the mesial cusp.
U6D Upper first molar distal tip	Most prominent point on the distal cusp.
L6M Lower first molar mesial tip	Most prominent point on the mesial cusp.
L6D Lower first molar distal tip	Most prominent point on the distal cusp.
SEM (Sphenoethmoidal point)	Intersection of the greater wing of sphenoid and the cranial floor.
Pn (Pronasale)	The most anterior point on the nose.
Co (Condylion)	The most superior and posterior point on the condylar head.

**Table 2 bioengineering-09-00617-t002:** Description of the model parameters.

Parameter	Description
Nu	Weight given to new trees being successively added to the present ensemble.
Oversampling Amount	Number of times the same image is used for different regressors.
Cascade Depth	Total number of cascade stages.
Tree Depth	Depth of each regression tree in the ensembles.

**Table 3 bioengineering-09-00617-t003:** Parameters used in each type of image transformation used for data augmentation.

Zoom	H-Shifted	W-Shifted	Shear	Rotate	Elastic
10%	50 pixels	50 pixels	10 deg	10 deg	alpha = 100, sigma = 5, alpha_affine = 50

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
