# Peer review of "Learning Cephalometric Landmarks for Diagnostic Features Using Regression Trees"

_bioengineering, 2022, doi:10.3390/bioengineering9110617_

Round 1
Reviewer 1 Report
The paper entitled “Learning Cephalometric Landmarks for Diagnostic Features Using Regression Trees” is a very interesting contribute that objective to automate the detection of cephalometric landmarks using machine learning and then compute diagnostic features using the landmark locations.
The work provides original data of absolute interest for clinicians but requires some corrections before it can be considered valid for publication.
INTRODUCTION
Overall well structured, it provides all the information necessary to understand the scientific background, the knowledge gap and the objectives of the study.
MATERIAL AND METHODS
All aspects relating to the scientific methodology used were described in a clear and exhaustive manner.
The design of the study, the methods and types of outcomes measured, the type of statistical analysis carried out for the interpretation of the data is correct and clearly reported.
RESULTS AND
The results are described in a precise and detailed manner; tabular and graphical representation is well executed and allows a faster understanding of the results achieved in the study.
DISCUSSION
The discussion of the results is on the whole well articulated ; clinical relevance of the results should be emphasized more.
Fig. 7 must be removed from the discussion section and inserted in the results
CONCLUSION
Conclusions are limited to a synthetic summary of the results obtained; this section must be revised and report preferably with a bulleted list, only the key results of the study.
Author Response
Thank you for the opportunity to revise our paper.
Below are the comments (C) and the response (A)
Reviewer 1
The paper entitled “Learning Cephalometric Landmarks for Diagnostic Features Using Regression Trees” is a very interesting contribution to automate the detection of cephalometric landmarks using machine learning and then compute diagnostic features using the landmark locations.
The work provides original data of absolute interest for clinicians but requires some corrections before it can be considered valid for publication.
C-1. Overall well structured, it provides all the information necessary to understand the scientific background, the knowledge gap and the objectives of the study.
A-1. Thank you for the comments.
C-2. All aspects relating to the scientific methodology used were described in a clear and exhaustive manner.
The design of the study, the methods and types of outcomes measured, the type of statistical analysis carried out for the interpretation of the data is correct and clearly reported.
A-2. Thank you for the comments.
C-3. The results are described in a precise and detailed manner; tabular and graphical representation is well executed and allows a faster understanding of the results achieved in the study.
A-3. Thank you for the comments.
C-4. The discussion of the results is on the whole well-articulated ; clinical relevance of the results should be emphasized more.
Fig. 7 must be removed from the discussion section and inserted in the results
A-4. Thank you for the comments. The changes have been made according to the reviewer’s comments. The discussion has been modified to include emphasize the clinical relevance of the results. Figure 7 has been moved to the results.
C-5. Conclusions are limited to a synthetic summary of the results obtained; this section must be revised and report preferably with a bulleted list, only the key results of the study.
A-5. Thank you for the comments. The changes have been made according to the reviewer’s comments. The conclusion has been revised with a bulleted list including only the key results of the study.
Once again , thank you for your valuable inputs! We appreciate your time and effort.
Reviewer 2 Report
The study is interesting and is based on the automatization and machine learning. The introduction is clear but it should be more focused on the limitation of cephalometric analysis. Material and methods are clear. The results and the discussion should be clarified better underlying the improvements that this method will bring to the cephalometric analysis.
Author Response
Thank you for your valuable inputs! We appreciate your time and effort.
Reviewer 2
Q-1. The study is interesting and is based on the automatization and machine learning. The introduction is clear but it should be more focused on the limitation of cephalometric analysis. Material and methods are clear. The results and the discussion should be clarified better underlying the improvements that this method will bring to the cephalometric analysis.
A-1. Thank you for the comments. The changes have been done according to the reviewer’s comments. The results and discussion have been modified to highlight the improvements that this method will bring to the cephalometric analysis.
The following changes have been made.
“When training a machine learning algorithm, increasing the number of training data samples results in a higher accuracy on unseen images. However, increasing the amount of labelled data is a challenge in the orthodontic field as labeling the cephalometric radiographs requires inputs from the orthodontist. This results in increased expert labeling costs and substantial expert time. Both these factors can limit the amount of labelled data available. Therefore, it is desirable to in-crease the size of the training set while also keeping the expert labeling costs of the x-rays to the minimum. This research study shows that augmentation methods can be used successfully to in-crease the training the data sample using an ensemble regression tree method for analysis of cephalometric radiographs.”
Once again , thank you for your valuable inputs! We appreciate your time and contribution.
Reviewer 3 Report
This article aimed to study to automate the detection of in cephalometric x-rays landmarks using machine learning and then compute diagnostic features using the landmark locations, and a number of data augmentation methods and their performance were evaluated. Although this study is interesting and well-written, further discussion should be necessary at following points.
1. The author said in discussion, “Specifically, the prediction errors for angle [A point subspinale]-[Nasion]-[B point supramentale] were within 4 degrees. We are therefore justified to conclude that the current model can be used by orthodontists in their practices for landmark identification.”. However, for orthodontists, 4 degrees is very big error in practice. If author would like to insist that that error is justified, please add the reasons and detailed explanation.
2. In discussion, the author said that “ANB measures the difference between SNA and SNB. The prediction of SNA, SNB & ANB showed some error as perthe current model (Figures 8) . However interestingly the angle ANB, demonstrated a smaller error than both SNA and SNB.” The smaller difference of ANB may show the error of each SNA and SNB per sample. Please add more discussion about this result.
Author Response
Thank you for the opportunity to revise our paper.
Below are the comments (C) and the response (A)
Reviewer 3
This article aimed to automate the detection of in cephalometric x-rays landmarks using machine learning and then compute diagnostic features using the landmark locations, and also evaluate a number of data augmentation methods and their performance. Although this study is interesting and well-written, further discussion should be necessary at following points.
C-1. The author said in discussion, “Specifically, the prediction errors for angle [A point subspinale]-[Nasion]-[B point supramentale] were within 4 degrees. We are therefore justified to conclude that the current model can be used by orthodontists in their practices for landmark identification.”. However, for orthodontists, 4 degrees is very big error in practice. If author would like to insist that that error is justified, please add the reasons and detailed explanation.
A-1. Thank you for the comments. The changes have been made according to the reviewer’s comments. We agree with the reviewer’s comments that 4 degrees is a significant error from a clinical standpoint. What we meant was that the standard deviation for ANB was within 3.6 degrees. A recent research in the field of AI published recently has shown that they have considered a standard deviation of 3.5 degrees acceptable regarding ANB. We have edited the discussion according to the reviewer’s comments.
“The current model not only predicts the individual landmarks with appreciable accuracy, but also preserves the angular relationships between these landmarks which is crucial for correct diagnosis (Figure 8). Specifically, the standard deviation of the errors for angle [A point subspinale]-[Nasion]-[B point supramentale] were within 4 degrees. This is similar to the research findings by Hunag et al. on the analysis of cephalometric measurements using artificial intelligence-based algorithms. [25]”
We have deleted the following:
“We are therefore justified to conclude that the current model can be used by orthodontists in their practices for landmark identification.”
C-2. In discussion, the author said that “ANB measures the difference between SNA and SNB. The prediction of SNA, SNB & ANB showed some error as per the current model (Figures 8). However interestingly the angle ANB, demonstrated a smaller error than both SNA and SNB.” The smaller difference of ANB may show the error of each SNA and SNB per sample. Please add more discussion about this result.
A-2. Thank you for the comments. The changes have been done according to the reviewer’s comments. The following has been added to the discussion
“ANB measures the difference between SNA and SNB. The prediction of SNA, SNB & ANB showed some error as per the current model (Figures 8). However interestingly the angle ANB, demonstrated a smaller error than both SNA and SNB. ANB is the net output obtained by the difference between SNA and SNB. The smaller difference of ANB may show the error of each SNA and SNB per sample. This could be due to the directionality of the errors in SNA and SNB such that the errors lined up antagonistically with each other and not synergistically resulting in a net lower ANB error.”
Thank you again for your valuable inputs! We appreciate your time and contribution.
Reviewer 4 Report
The topic is interesting.
1. Authors should describe the results in the corresponding section. Thus, figures 7 and 8 as well as their description should be placed in the results section.
2. The discussion should be enhanced with results from other similar studies on literature and with reference to the limitations of this study.
3. Conclusions should be more objective.
Author Response
Thank you for the opportunity to revise our paper.
Below are the comments (C) and the response (A)
Reviewer 4
The topic is interesting.
C-1. Authors should describe the results in the corresponding section. Thus, figures 7 and 8 as well as their description should be placed in the results section.
A-1. Thank you for the comments. The changes have been done according to the reviewer’s comments. The figure and the description have been placed in the results section
C-2. The discussion should be enhanced with results from other similar studies on literature and with reference to the limitations of this study.
A-2. Thank you for the comments. The changes have been done according to the reviewer’s comments. The discussion has been modified extensively. Other studies from the literature and the limitation of the study have also been added to the discussion as per the reviewer’s comments.
C-3. Conclusions should be more objective.
A-3. Thank you for the comments. The changes have been done according to the reviewer’s comments. The conclusion has been modified in a bullet format with specific conclusions.
Once again, thank you for your valuable inputs! We appreciate your time and effort.
Round 2
Reviewer 3 Report
The authors have addressed all critiques raised by my comments.